# Probiotic Properties Including the Antioxidant and Hypoglycemic Ability of Lactic Acid Bacteria from Fermented Grains of Chinese Baijiu

**DOI:** 10.3390/foods11213476

**Published:** 2022-11-02

**Authors:** Sanhong Fan, Tengda Xue, Baoqing Bai, Tao Bo, Jinhua Zhang

**Affiliations:** 1College of Life Sciences, Shanxi University, Taiyuan 030006, China; 2Shanxi Key Laboratory of Research and Utilization of Characteristic Plant Resources, Shanxi University, Taiyuan 030006, China

**Keywords:** fermented grains, Baijiu, lactic acid bacteria, probiotic properties

## Abstract

In this study, lactic acid bacteria (LAB) strains were isolated from fermented grains of traditional Chinese Baijiu, and their probiotic properties were characterized. Eleven out of 29 LAB strains showed good tolerance to the gastrointestinal tract and bile salts. The surface characteristics (auto-aggregation, co-aggregation, hydrophobicity), safety (hemolytic and antibiotic sensitivity), antibacterial activity against three foodborne pathogens, and antioxidant and hypoglycemic properties of the 11 LAB strains were investigated. Principal component analysis (PCA) was used to comprehensively evaluate LAB strains and their probiotic properties. It was found that *Weissella cibaria* (OP288150), *Pediococcus acidilactici* (OP288151), *Pediococcus pentosaceus* (OP288154), *Pediococcus pentosaceus* (OP288156) and *Levilactobacillus brevis* (OP288158) showed high probiotic properties, with potential for commercial development. The results also demonstrated that fermented grains of Chinese Baijiu can be used as a source of high-quality probiotics.

## 1. Introduction

Probiotics are defined by the Food and Agriculture Organization of the United Nations (FAO) and the World Health Organization (WHO) as “live microorganisms, which when administered in adequate amounts confer a health benefit on the host” [1]. First, probiotics colonized in the human intestine produce various nutrients for the host, maintain the ecological balance of the intestinal tract, regulate the normal immune response, and prevent a variety of diseases [2]. They have specific anticancer potential [3]. In addition, probiotics, mainly yeast and lactic acid bacteria (LAB) [4], play a crucial role in the preservation and transformation of food [5]. They are involved in the fermentation of foods closely related to human history and civilization, including fermented milk, wine, vinegar, bread, pickles, and soy sauce [6].

In recent years, probiotics have been utilized in the development and utilization of functional foods, dietary supplements, and biopharmaceuticals [7,8]. Numerous studies have shown that foods containing probiotics, such as dairy and soybean products, are easily absorbed by the body, with high nutritional value, and have health benefits [9,10]. Supplementation with a combination of probiotics accelerates the maturation of gut microbiome maturation and reduces intestinal inflammation in extremely preterm infants [11]. In addition, probiotics metabolize a variety of bioactive substances during fermentation of food, which contribute to human health [12,13]. Currently, as functional foods or food supplements, probiotics are widely available in the market for public consumption. They play a key role in food science and clinical medicine, highlighting the need to develop high-quality probiotics and related probiotics products [14,15,16]. 

Fermented foods are the most common natural source of potential probiotics [17]. Chinese Baijiu is one of the oldest fermented foods. It is a traditional distilled wine obtained by spontaneous fermentation of fermented grains (containing a mixture of Daqu and Sorghum in specific proportions) under specific environments [18]. During the fermentation process, the fermented grains and the microorganisms develop under a similar environment. During the early stage of fermentation, molds are the main microorganisms in aerobic and highly humid environments. In the early-to-middle fermentation stage, yeast is the main species. The low oxygen, low pH and high alcohol content during the late stage of fermentation result in LAB growth predominantly [18,19,20]. The unique environment of fermented grains at different stages contributes to the resistance of LAB and yeast and their potential utilization as probiotics.

In this study, LAB were isolated from fermented grains at different stages of fermentation. The strains with strong tolerance were further screened based on acid resistance, gastrointestinal tract tolerance and bile tolerance. The potential probiotic ability of LAB was evaluated according to the surface characteristics (auto-aggregation, co-aggregation, and hydrophobicity), safety (hemolytic activity and antibiotic sensitivity), antibacterial activity, antioxidant, and hypoglycemic ability, in an effort to isolate potent probiotics from fermented grains of Chinese Baijiu. In addition to their performance as probiotic candidates, the selected LAB isolates can be utilized in other fermented foods such as kimchi, milk, and soymilk.

## 2. Materials and Methods

### 2.1. Sampling and Isolation of LAB

Fermented grains were collected from Shanxi Xinghuacun Fenjiu Co., LTD. (Fenyang, Shanxi, China). Fermented grains obtained via fermentation times for 0, 7, 15 and 28 days in spring were transferred into sterile plastic bags and transported to the laboratory of Shanxi University under dry ice. Each sample (5 g) was transferred to 45 mL phosphate buffer solution (PBS, pH 7.4), incubated at 37 °C for 10 min, gradient diluted and inoculated on MRS agar (Solarbio, Beijing, China), and incubated at 37 °C for 24–72 h. The culture medium of the isolated strain was mixed with 30% glycerol solution in equal volumes and stored at −50 °C.

### 2.2. Preparation of the Cell-Free Supernatant (CFS) and Intact Cells (IC)

LAB strains metabolize extracellular active compounds during the growth, which are then released into the supernatant following centrifugation. The surface agents and intracellular products of intact bacterial cells also exhibit certain functions. The cell-free supernatant and intact cells were obtained to comprehensively evaluate the probiotic potential of isolated LAB strains.

LAB strains were incubated overnight in MRS broth (Solarbio, Beijing, China) under aerobic conditions at 37 °C, centrifuged at 12,000 rpm for 8 min at 4 °C, and filtered through a hydrophilic membrane with a pore size of 0.22 μm to obtain CFS. 

The cells obtained by centrifugation, as described above, were washed three times with sterile phosphate buffer solution (PBS, pH 7.4) and resuspended in an equal volume of PBS to prepare bacterial suspension. The thallus concentration was adjusted to 1.0 × 108 CFU/mL (OD600 nm = 1.0).

### 2.3. Tolerance to pH

Tolerance to different pH levels was evaluated as described by Mishra et al. [21] with some modifications. Briefly, LAB strains were incubated in MRS broth at 37 °C for 12 h and activated for 3 generations. The culture medium with a volume fraction of 1% was inoculated into MRS liquid medium with pH 3.5 and 3.0 adjusted by 2 mol/L HCl, then incubated at 37 °C for 24 h. The visible turbidity of the culture was observed after shaking. LAB strains that make the visible turbidity were selected to test for tolerance. 

### 2.4. Tolerance to Gastric and Intestinal Juice

The tolerance of the isolated strains to artificial gastric and intestinal juice was determined as described by Son et al. [22]. Briefly, an overnight culture of the LAB strains was resuspended in the MRS broth, and 1 mL of an activated culture was mixed with 9 mL simulated artificial gastric juice (1 g/100 mL pepsin, pH 2.0 and intestinal juice (1 g/100 mL pancreatin, 0.68 g/mL KH_2_PO_4_, pH 8.0), followed by incubation at 37 °C. After 0 h and 2 h, samples were removed and gradient diluted, and then coated on an MRS agar plate. After 24 h of incubation at 37 °C, the plates were counted using a colony counter. 

### 2.5. Bile Tolerance

Bile tolerance of the isolated strains was tested against bile salt (0.3% and 1%) and sodium taurocholate (1%), according to Tarique et al. [23]. The absorbance was measured at 620 nm at 0, 3, and 6 h. The percentage of bile tolerance was estimated using the following equation:$$\%\;\mathrm{of}\;\mathrm{bile}\;\mathrm{tolerance}=\left[\left(A_{control}-A_{bile}\right)/A_{control}\right]\times 100$$
where *A_control_* represents the absorbance of the bacterial growth in MRS broth alone, and *A_bile_* indicates the absorbance of bacterial growth in MRS broth supplemented with bile salt.

### 2.6. Auto-Aggregation

Auto-aggregation of the isolated strains was detected as described by Xu et al. [24]. Briefly, cells grown overnight were suspended in PBS buffer. The LAB suspension (4 mL) was vortexed for 10 s and incubated at 30 °C. Samples were monitored during different time intervals (0 h and 4 h), and the absorbance was measured at 600 nm. The percentage of auto-aggregation was calculated based on the following equation:$$\mathrm{Auto}-\mathrm{aggregation}\;\left(\%\right)=\left[1-(A_{t}/A_{0}\right)]\times 100$$
where *A_t_* stands for absorbance at time “*t*” and *A*_0_ represents absorbance at *t* = 0.

### 2.7. Co-Aggregation

Co-aggregation of the isolated strains was detected as described by Ayyash et al. [25] using three foodborne pathogenic bacteria (*E. coli*, *Staph. aureus*, *S. typhi*). Briefly, the LAB suspensions were mixed with the same volume of pathogenic bacterial suspensions, and the suspensions were vortexed for 10 s before standing. Samples were monitored during different time intervals (0 h and 4 h), and the absorbance was measured at 600 nm. Co-aggregation percentage was calculated based on the following equation:$$\mathrm{Co}-\mathrm{aggregation}\;\left(\%\right)=\left[1-(A_{t}/A_{0}\right)]\times 100,$$
where *A_t_* represents absorbance at time “*t*” and *A*_0_ refers to absorbance at *t* = 0.

### 2.8. Hydrophobicity

The hydrophobicity of the isolated strains was determined by measuring their adhesion to different hydrocarbons (chloroform, ethyl acetate or hexane), according to the procedure described by Munoz-Provencio et al. [26]. Briefly, 3 mL of LAB suspension was mixed with 1 mL of different solvents. The mixture was vortexed for 1 min and allowed to stand for 4 h, and the absorbance was measured at 600 nm. The percentage of hydrophobicity was calculated based on the following equation:$$\mathrm{Hydrophobicity}\;\left(\%\right)=\left[1-\left(A_{t}/A_{0}\right)\right]\times 100$$
where *A_t_* is the absorbance at time “*t*” and *A*_0_ represents the absorbance at *t* = 0.

### 2.9. Safety Profile

#### 2.9.1. Hemolytic Activity

Hemolytic activity was determined as described by Angmo et al. [27]. Briefly, LAB strains were seeded on the surface of Columbia blood agar (CBA) supplemented with 5% (*v/v*) sterile defibrinated sheep blood and cultured at 37 °C for 24–48 h. *Staph. aureus* was used as a positive control to observe a clear circle.

#### 2.9.2. Antibiotic Susceptibility Test

Disk diffusion was performed to test for antibiotic susceptibility according to Xu et al. [24]. Briefly, the LAB cell suspensions were spread on MRS agar plates. Drug susceptibility test paper (Bkmam, Changde, China) impregnated with different antibiotics was placed on the surfaces. In this study, the sensitivity of the isolated strains was tested against ten antibiotics, including chloramphenicol (C, 30 μg), vancomycin (VAN, 30 μg), streptomycin (S, 10 μg), penicillin (PEN, 10 μg), clindamycin (CC, 2 μg), piperacillin (PIP, 100 μg), oxacillin (OX, 1 μg), cefazolin (CZ, 30 μg), trimethoprim-sulfamethoxazole (SXT, 25 μg), and minocycline (MT, 30 μg). 

### 2.10. Antibacterial Activity

The antibacterial activity of the isolated strains against some foodborne pathogenic bacteria was evaluated using the Oxford cup method described by Wang et al. [28]. Briefly, the three strains of pathogenic bacteria (*E. coli*, *Staph. aureus*, and *S. typhi*) were diluted to a concentration of 107–108 CFU/mL (OD600 = 0.5 ± 0.05). A 0.1 mL aliquot of bacterial solution was evenly spread on the surface of LB agar. The center of the Oxford cup (Φ 7.8 mm) was placed on the surface of medium symmetrically, and 100 μL CFS was injected into each Oxford cup. The plate was incubated at 37 °C for 24 h, and the diameter of the inhibition zone was observed and measured.

### 2.11. Antioxidant Ability

DPPH radical scavenging activity was measured according to Wang et al. [29]. Briefly, 1 mL sample (CFS or IC) was mixed with 1 mL DPPH solution (0.2 mmol/L, dissolved in absolute ethanol), reacted at 37 °C in the dark for 30 min, and centrifuged at 6000 r/min for 10 min. The supernatant was used to measure the absorbance at 517 nm. The reducing activity of ascorbic acid was used as a positive control. The blank was treated with an equal volume of absolute ethanol solution instead of DPPH absolute ethanol. The control was treated with an equal volume of distilled water instead of sample solution. DPPH radical scavenging activity was calculated as follows:$$\mathrm{DPPH}\;\mathrm{radical}\;\mathrm{scavenging}\;\mathrm{activity}\;\left(\%\right)=[1-\left(A_{sample}-A_{blank})/A_{control}\right]\times 100$$
where *A_sample_* represents the absorbance of sample, *A_blank_* denotes the absorbance of the blank, and *A_control_* is the absorbance of the control. 

### 2.12. Hypoglycemic Ability

#### 2.12.1. α-Amylase Inhibition Activity

The inhibition of α-amylase was evaluated as described by Lee et al. [30]. Briefly, 500 µL samples (CFS or IC) were mixed with PBS (500 µL, pH 7.4) and α-amylase solution (500 µL, 0.5 mg/mL) and incubated at 37 °C for 10 min, followed by the addition of 500 µL of 1% starch solution (*w/v*) and incubated at 37 °C for 3 min. Followed by the addition of 1 mL DNS solution (3,5-dinitrosalicylic acid), the solution was incubated in a boiling water bath for 5 min. Next, 5 mL distilled water was added and cooled to room temperature, and the absorbance was measured at 540 nm. Acarbose was used as a positive control.
$$\alpha -\mathrm{amylase}\;\mathrm{inhibition}\;\mathrm{rate}\;\left(\%\right)=\left[1-(A_{sample}-A_{blank})/A_{control}\right]\times 100$$
where *A_sample_* refers to absorbance of the experimental sample (samples and α-amylase), *A_blank_* indicates the absorbance of the blank (samples without α-amylase), and *A_control_* represents the absorbance of the control (with α-amylase and without samples).

#### 2.12.2. α-Glucosidase Inhibitory Activity

The inhibition of α-glucosidase was evaluated as described by Li et al. [31]. Briefly, 25 µL sample, 50 µL glucosidase solution (0.2 U/mL), 150 µL PBS (0.1 mol/L, pH 7.4), and 75 µL p-nitrobenzene-α-D-glucoside (pNPG) solution (dissolved in PBS, 20 mmol/L) were mixed. After incubation at 37 °C for 10 min, the reaction was terminated by adding 1 mL of 0.1 mol/L Na_2_CO_3_ solution, and the absorbance was measured at a wavelength of 405 nm. Acarbose was used as a positive control. The α-glucosidase inhibition was evaluated using the following formula:$$\alpha -\mathrm{glucosidase}\;\mathrm{inhibition}\;\mathrm{rate}\;\left(\%\right)=\left[1-(A_{sample}-A_{blank}/A_{control}-A_{0}\right)]100$$
where *A_sample_* is the absorbance of the experimental sample (with samples and α-glucosidase), *A_blank_* denotes the absorbance of the blank (with samples but without α-glucosidase), *A_control_* is the absorbance of the control (with α-glucosidase and without samples), and *A*_0_ represents the absorbance without α-glucosidase and samples. 

### 2.13. Identification of LAB Strains

Bacterial DNA extraction kit (Bio Teke, Beijing, China) was used to extract the genomic DNA of LAB strains, and PCR amplification was performed. The 16S rRNA gene was amplified using universal primers 27f (5′-AGAGTTTGATCCTGGCTCAG-3′) and 1492r (5′-ACGGCTACCTTGTTACGACTT-3′) [32]. PCR was performed using a 25 μL amplification reaction system: 1.0 μL DNA template (2.0 ng/μL), 2.5 μL upstream and downstream primers (10 μmol/L), 12.5 μL 2 × Taq PCR Master mix, supplemented with ultrapure water to 25 μL. PCR amplification conditions were: initial denaturation at 94 °C for 5 min, denaturation at 94 °C for 30 s, annealing at 55 °C for 30 s, extension at 72 °C for 1 min, 35 cycles, and final elongation at 12 °C. PCR products were detected by 2% gel electrophoresis and sequenced.

### 2.14. Statistical Analysis

All the above experiments were repeated three times. Data are presented as mean ± standard deviation. Statistical analyses were performed using SPSS 22.0 (IBM^®®^ SPSS^®®^ Statistics, Chicago, IL, USA), and *p* < 0.05 was considered statistically significant. Principal component analysis (PCA) of LAB and probiotic properties was performed using Origin 2018.

## 3. Results and Discussion

### 3.1. Screening of Acid-Resistant LAB Strains

A total of 29 LAB strains was selected from four fermented grains samples based on colony characteristics, Gram staining and catalase activity experiments. Acid resistance determines whether the strain can grow normally in foods with high acidity (such as yogurt). Further, acid resistance determines whether LAB strains can enter the human gastrointestinal tract and play a probiotic role. The pH value of human gastric juice is usually about 3.0 [33]. Therefore, in this experiment, pH levels of 3.5 and 3.0 were used to screen acid-resistant LAB strains and determine their acid tolerance based on visible turbidity of the culture. The results are shown in Table 1. All 29 strains could survive at pH 3.5, but only 19 strains could survive at pH 3.0. Next, the tolerance ability of acid-resistant strains was further analyzed.

### 3.2. Tolerance to Gastric and Intestinal Juice

The above acid-tolerant LAB strains were inoculated into a simulated gastrointestinal environment to determine their tolerance to gastric and intestinal fluids. The results are shown in Table 2. The average LAB counts in activated culture ranged from 7.76 to 9.29 Log10 CFU/mL and were inoculated into gastric juice (pH 2.0) and intestinal juice (pH 8.0) at 10% (*v/v*), respectively. After 2 h in gastric juice, 11 surviving strains remained with colony counts ranging from 2.90 to 7.80 Log10 CFU/mL. After 2 h in the intestinal juice, except JP1507 and JP1512, the other strains survived in the intestinal juice and showed higher tolerance, with an average colony count ranging from 6.36 to 8.69 Log10 CFU/mL. The results indicated that LAB had better tolerance ability. A similar trend reduction was reported previously [25,34,35] when strains were exposed to low pH gastric and intestinal juices. 

Adequate bacterial activity during gastrointestinal transport is a prerequisite for the health benefits of probiotics entering the human body [36]. Tolerance to gastrointestinal conditions (low pH, digestive enzymes, bile salts) is essential for probiotic activity [23,37]. Accordingly, 11 strains that survived in both gastric juice and intestinal juices were selected for further analysis of their potential probiotic properties.

### 3.3. Bile Salt Tolerance

Bile salts are toxic to living cells, and tolerance to bile salts is one of the properties required for the survival of LAB in the small intestine [38,39]. The percentages of growth of 11 LAB strains treated with bile salts (0.3% and 1%) and sodium taurocholate (1%) are shown in Table 3. A higher percentage of growth indicates good bile salt tolerance [40]. As shown in Table 3, strains JP002, JP7(02, 03, 04, 05, 10) and JP15(02, 10) showed resistance to bile salts. All strains showed resistance to sodium taurocholate. In general, the tolerance to sodium taurocholate was higher than that of bile salts. These results were similar to studies with strains isolated from dairy products, with LAB strains showing lower tolerance to cholic acid, and ox gall than taurocholic acid [23,40].

### 3.4. Auto-Aggregation and Co-Aggregation

Percentages of auto-aggregation and co-aggregation of the three study pathogens (*E. coli*, *Staph. aureus*, *S. typhi*) of LAB strains are listed in Table 4. Auto-aggregation is related to the strain’s ability to adhere and colonize the gastrointestinal tract. Our LAB strains showed a good percentage of auto-aggregation ranging from 2.11 to 27.77% during 4 h of incubation. This result is similar to the results of Abushelaibi et al. and Angmo et al. [27,40], but lower than the auto-aggregation of LAB strains reported by Sakoui et al. [41]. 

In addition, the co-aggregation of probiotic strains with pathogens promotes adherence to pathogenic bacteria and their removal [41]. The percentage of co-aggregation of our strain with the three foodborne pathogens ranged from 7.09% to 77.61%. Different strains exhibit different levels of copolymerization in the presence of different pathogenic bacteria, and the co-aggregation percentage of *S. typhi* was higher than that of the other strains. The mechanism of inhibition of pathogenic bacteria by probiotics in the human gut was proposed by Shipradeep et al. [42]. Co-aggregation of probiotics and pathogenic microorganisms is considered to be the most important mechanism. 

### 3.5. Hydrophobicity

The surface hydrophobicity of probiotics is related to probiotic attachment to host tissues, which plays a vital role in various biological interactions [43,44]. Table 5 shows the hydrophobicity of different hydrocarbons (chloroform, ethyl acetate or hexane). The strains exhibited hydrophobicity of 14.43–74.62%, 15.71–70.31%, and 6.71–88.44% for chloroform, ethyl acetate and hexane, respectively. Different strains showed different levels of hydrophobicity against different hydrocarbons. The bacterial surface was considered hydrophobic when the hydrophobic index was greater than 70% [45]. Therefore, based on our results, JP1502 is hydrophobic to chloroform; JP001 is hydrophobic to ethyl acetate, and JP703 and JP705 are hydrophobic to hexane. The results differ from those of Sakoui et al. [41], where chloroform was the most hydrophobic, followed by hexane and ethyl acetate. 

### 3.6. Safety Profile

The premise of probiotics application is that they are not pathogenic to humans. Microorganisms that produce hemolysin, which can dissolve red blood cells, are toxic to the human body; probiotics with antibiotic resistance may transfer associated resistance genes to microorganisms in the gastrointestinal tract [46]. The safety of the selected LAB strains was tested based on hemolytic and antibiotic sensitivity, and the results are shown in Figure 1 and Table 6. None of the strains produced precise circles on Columbia blood agar and were negative for hemolytic activity. The strains JP704 and JP708 were susceptible to all antibiotics, while the other strains showed vancomycin resistance. However, all strains were susceptible to chloramphenicol, streptomycin, minocycline, and the effects of other antibiotics on the strains ranged from sensitive to inhibitory, which was similar to the results reported by Nami et al. [47]. In the study of Abushelaibi et al. [40], strains isolated from camel milk were resistant to vancomycin and trimethoprim, but susceptible to penicillin and clindamycin. The differences in the present study may be due to the variety of strains. According to previous studies [23,37,48], some probiotics are naturally resistant to common antibiotics, such as vancomycin. Resistance to antibiotics does not determine the safety of strains, and their resistance may also be intrinsic and not induced by infections [24]. Nevertheless, the hemolytic test demonstrated the good safety profile of the isolated strains.

### 3.7. Antibacterial Activity

Potential probiotics should exhibit antibacterial activity and contribute to the health of the gastrointestinal tract by inhibiting the growth and reproduction of pathogenic bacteria [49,50]. In this study, the cell-free supernatant of the strains was used to determine their antibacterial ability of the strains against *E. coli*, *S. typhi* and *Staph. aureus*. The results of the size of the inhibition zone are shown in Table 7. All strains showed inhibition against the three pathogenic bacteria, with more inhibition against *S. typhi* than against *E. coli* and *Staph. aureus*. In relative terms, strain JP1510 strongly inhibited all three tested pathogens. Studies have shown that LAB inhibit the growth of pathogenic bacteria via competitive inhibition and metabolism of bacteriostatic extracellular products. However, the strength of antimicrobial activity depends on species and strains, and different species and different isolates exhibit different bacteriostatic activity [51,52,53].

### 3.8. Antioxidant Activity

Oxidative stress can induce harmful cellular effects. Probiotics promote antioxidant effects by strengthening the body’s antioxidant response and reducing free radical production [54]. Further, probiotics with antioxidant activity reduce the risk of ROS accumulation during dietary intake and help prevent various diseases [55,56]. Probiotics with antioxidant activity have higher potential for application.

This study investigated the antioxidant activities of CFS and IC of 11 LAB strains via DPPH radical scavenging activity assays. As shown in Figure 2, the average scavenging rates of CFS and IC of LAB strains for DPPH free radicals ranged from 63.65% to 89.61% and 35.59% to 47.75%, respectively, and the antioxidant activity of 0.01 mg/mL ascorbic acid was 76.30%. The CFS of strains JP701, 704, 705 and 710 exhibited good scavenging activity, while the IC of strain JP703 showed the highest scavenging activity. In general, the screened LAB strains showed similar DPPH radical scavenging activity. The antioxidant capacity of IC was significantly lower than that of CFS, which was consistent with the results of Tang et al. [57]. In one study, the DPPH clearance rates of CFS and IC of LAB strains were 57.43–73.02% and 4.50–19.66%, respectively [29], which were slightly lower than those reported in this study.

### 3.9. Hypoglycemic Ability

The blood glucose level in the body is closely related to the activities of α-amylase and α-glucosidase. Therefore, inhibiting these enzyme activities can control the absorption of carbohydrates and effectively reduce blood glucose levels, thereby inhibiting chronic diseases such as type 2 diabetes mellitus and other chronic diseases induced by hyperglycemia [30]. In this study, the hypoglycemic activity of 11 LAB strains was evaluated by measuring the inhibitory rates of α-amylase and α-glucosidase.

As shown in Table 8, the inhibitory activity was detected in CFS and IC of LAB. The average ranges of α-amylase inhibition rates of CFS and IC of 11 LAB strains were 17.19–86.15% and 29.11–72.38%, respectively. The CFS of JP701 showed the highest α-amylase inhibition activity (86.15%), while the CFS of JP003 had the lowest α-amylase inhibitory activity (17.19%). However, the α-amylase inhibition by the IC of JP003 was the highest (72.38%). In general, strains JP701 and 702 exhibited strong α-amylase inhibitory activity. In the study by Wang et al. [29], LAB CFS showed excellent α-amylase inhibitory activity, while IC showed no apparent α-amylase inhibition. In contrast, in this study, both CFS and IC of LAB showed good inhibitory activity.

The inhibitory activity of α-glucosidase is shown in Table 8. The CFS and IC of 11 LAB strains exhibited appropriate inhibitory activities against α-glucosidase. The average inhibition rates were 45.31–60.86% and 43.96–54.61%, respectively, which were lower than those of 1 mg/mL acarbose (74.23%). The CFS of JP003 had the highest α-glucosidase inhibitory activity (61.74%), and the IC of JP718 had the highest α-glucosidase inhibitory activity (54.61%). In the study by Wang et al. [29], the inhibitory activities of CFS and IC of LAB strains were 29.04–85.16% and 73.81–83.81%, respectively, which were higher than in our study.

### 3.10. Identification of Selected LAB Strains by 16S rRNA

Eleven isolated strains were identified via 16S rRNA gene sequencing, and the species names and GenBank accession numbers are listed in Table 9. Specifically, seven LAB strains were identified, including one strain of *Weissella cibaria*, one strain of *Pediococcus acidilactici*, one strain of *Lactiplantibacillus plantarum* subsp. *Plantarum*, two strains of *Pediococcus pentosaceus*, two strains of *Levilactobacillus brevis*, one strain of *Leuconostoc mesenteroides*, and three strains of *Lentilactobacillus parabuchneri*. According to the 16S rRNA sequences obtained, the neighborhood joining method was used to construct the phylogenetic tree, as shown in Figure 3.

### 3.11. Principal Component Analysis (PCA)

PCA entails multivariate statistical analysis for screening of functional probiotics. In this study, PCA was used to screen high-quality strains among the 11 LAB strains. The results are shown in Figure 4. The results of PCA involving 11 LAB strains and 16 variables are shown in Figure 4. The first six principal components accounted for 89.6%, and the PC1 and PC2 were the most representatives, accounting for 25.2% and 19.2%, respectively. Significant correlations were found between the strains in quadrants 1 and 2 and the variables [41]. *W. cibaria* (OP288150), *P. acidilactici* (OP288151), *P. pentosaceus* (OP288154), *P. pentosaceus* (OP288156) and *L. brevis* (OP288158) were the probiotic candidates with the most potential.

## 4. Conclusions

The fermented grains of Chinese Baijiu were likely to be a reservoir of probiotic strains. In this study, LAB strains were isolated from fermented grains of Baijiu. The tolerance, surface properties, antioxidant activity and hypoglycemic ability of the isolated strains were comprehensively evaluated via PCA. *W. cibaria* (OP288150), *P. acidilactici* (OP288151), *P. pentosaceus* (OP288154), *P. pentosaceus* (OP288156) and *L. brevis* (OP288158) exhibited good probiotic potential and safety profile. They are potential probiotics for the production of functional foods and microbial preparations. The in vitro studies revealed the probiotic properties of the screened LAB strains. However, whether they can be utilized as probiotics for human consumption requires further studies in vivo to determine their safety and specific probiotic capacity.

## Figures and Tables

**Figure 1 foods-11-03476-f001:**
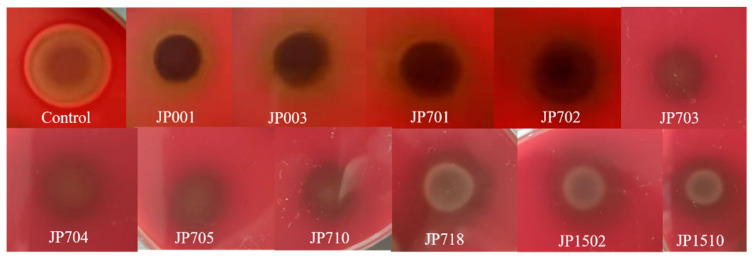
Hemolytic test of LAB strains. Hemolytic test results for *S. aureus* (Control) with transparent rings. None of the LAB strains produced circles of transparency.

**Figure 2 foods-11-03476-f002:**
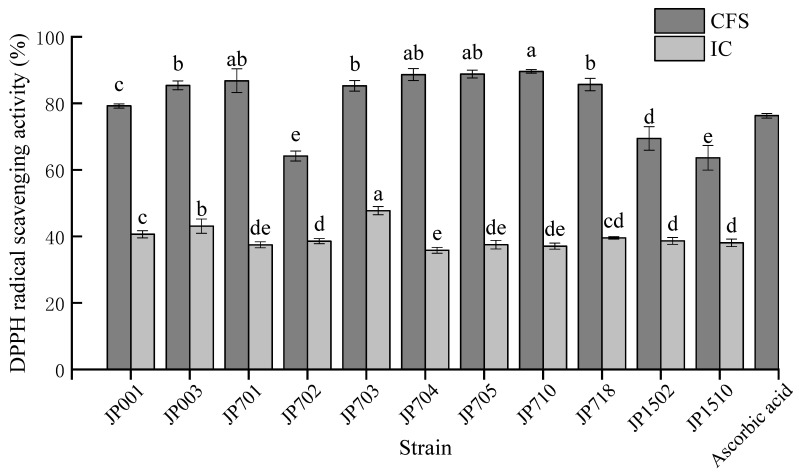
Antioxidant activity of the 5 LAB strains. Values are the mean ± standard deviation of triplicates. ^a–e^ Mean values in same column with different lowercase letters differ significantly (*p* < 0.05).

**Figure 3 foods-11-03476-f003:**
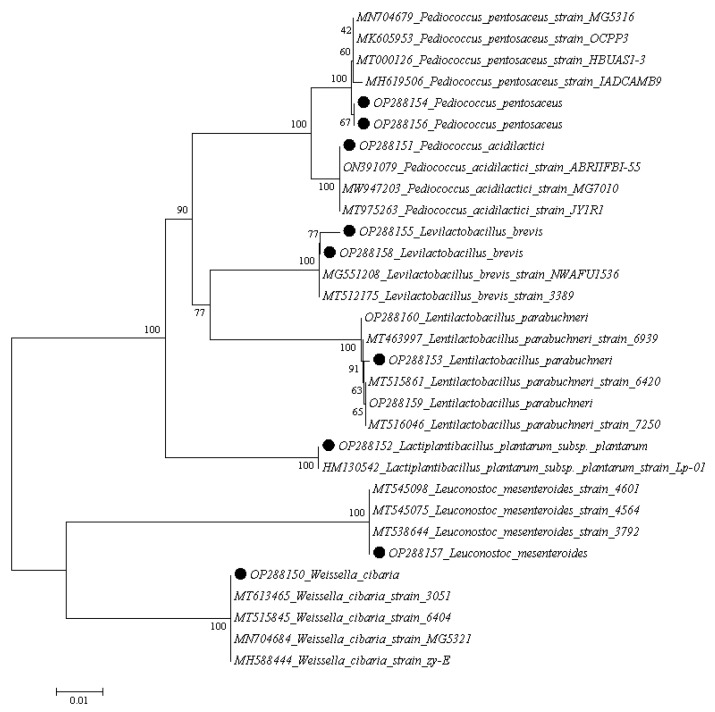
Phylogenetic tree based on 16S rRNA gene sequences analysis. Solid circles represent strains isolated from fermented grains of Baijiu.

**Figure 4 foods-11-03476-f004:**
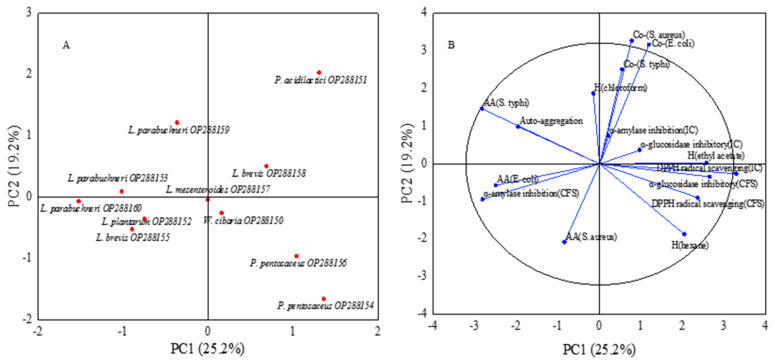
Principal component analysis (PCA) of 11 LAB strains and probiotic properties: (**A**) Projection of the LAB strains in the space of PC1 and PC2; (**B**) Projection of the probiotic properties on the plane formed by PC1 and PC2 analyzed by PCA.

**Table 1 foods-11-03476-t001:** Screening of acid-resistant strains of 29 LAB strains.

Strain	pH 3.5	pH 3.0	Strain	pH 3.5	pH 3.0
JP001	+	+	JP1506	+	+
JP003	+	+	JP1507	+	+
JP701	+	+	JP1508	+	+
JP702	+	+	JP1510	+	+
JP703	+	+	JP1511	+	−
JP704	+	+	JP1512	+	+
JP705	+	+	JP1513	+	+
JP710	+	+	JP1514	+	−
JP718	+	+	JP1515	+	+
JP719	+	−	JP1516	+	−
JP1501	+	−	JP2804	+	+
JP1502	+	+	JP2806	+	+
JP1503	+	−	JP2812	+	−
JP1504	+	−	JP2815	+	−
JP1505	+	−			

Note: “+”: Visible turbidity in the culture; “−”: No visible turbidity in the culture.

**Table 2 foods-11-03476-t002:** Tolerance to gastric and intestinal juice of 19 LAB strains (Log10 CFU/mL).

Strain	Activated Culture	Gastrointestinal Steps	
		After Gastric	After Intestine
JP001	8.82 ± 0.32 ^bc^	3.57 ± 0.51 ^d^	7.95 ± 0.17 ^bc^
JP003	9.00 ± 0.14 ^b^	6.19 ± 0.17 ^b^	7.86 ± 0.19 ^bc^
JP701	9.04 ± 0.18 ^b^	7.54 ± 0.16 ^a^	7.91 ± 0.23 ^bc^
JP702	9.59 ± 0.14 ^a^	7.80 ± 0.27 ^a^	8.26 ± 0.31 ^ab^
JP703	9.03 ± 0.12 ^b^	3.67 ± 0.35 ^d^	8.11 ± 0.22 ^b^
JP704	8.89 ± 0.17 ^bc^	5.06 ± 0.39 ^c^	8.27 ± 0.40 ^ab^
JP705	9.11 ± 0.04 ^ab^	5.77 ± 0.60 ^bc^	8.69 ± 0.06 ^a^
JP710	9.17 ± 0.08 ^ab^	2.90 ± 0.85 ^d^	8.47 ± 0.28 ^ab^
JP718	9.25 ± 0.61 ^ab^	3.67 ± 0.16 ^d^	7.92 ± 0.15 ^bc^
JP1502	7.76 ± 0.41 ^d^	4.88 ± 0.82 ^c^	7.54 ± 0.09 ^c^
JP1506	8.21 ± 0.52 ^cd^	ND	6.90 ± 0.27 ^d^
JP1507	8.23 ± 0.70 ^cd^	ND	ND
JP1508	8.18 ± 0.10 ^cd^	ND	7.53 ± 0.85 ^c^
JP1510	8.47 ± 0.17 ^c^	4.58 ± 0.06 ^c^	6.36 ± 0.36 ^e^
JP1512	7.94 ± 0.36 ^d^	ND	ND
JP1513	8.39 ± 0.12 ^cd^	ND	8.60 ± 0.27 ^ab^
JP1515	9.29 ± 0.11 ^ab^	ND	7.75 ± 0.13 ^bc^
JP2804	8.82 ± 0.12 ^bc^	ND	7.53 ± 0.22 ^c^
JP2806	8.60 ± 0.17 ^bc^	ND	8.31 ± 0.20 ^ab^

Note: Values are the mean ± standard deviation of triplicates. ND = Not detected (lower than 1 Log10 CFU/mL). ^a–e^ Mean values in same column with different lowercase letters differ significantly (*p* < 0.05).

**Table 3 foods-11-03476-t003:** Bile tolerances (%) of 11 LAB strains.

Strain	0.3% Bile Salt		1% Bile Salt		1% Sodium Taurocholate	
	3 h	6 h	3 h	6 h	3 h	6 h
JP001	15.74 ± 1.88 ^f^	75.88 ± 0.80 ^b^	23.71 ± 1.86 ^f^	70.91 ± 0.98 ^c^	23.13 ± 2.26 ^e^	76.66 ± 0.62 ^c^
JP003	62.17 ± 0.99 ^a^	81.36 ± 0.65 ^a^	62.79 ± 1.02 ^b^	62.09 ± 1.05 ^d^	63.61 ± 0.91 ^a^	83.03 ± 0.58 ^a^
JP701	4.39 ± 2.19 ^g^	15.29 ± 3.58 ^g^	2.67 ± 2.03 ^g^	32.54 ± 3.66 ^f^	5.94 ± 2.00 ^f^	46.32 ± 2.00 ^e^
JP702	51.33 ± 2.31 ^b^	71.43 ± 0.54 ^c^	51.63 ± 1.2 ^cd^	6.05 ± 1.83 ^g^	51.66 ± 1.21 ^bc^	81.22 ± 0.34 ^b^
JP703	44.3 ± 1.78 ^cd^	59.63 ± 1.16 ^e^	41.99 ± 1.66 ^e^	4.45 ± 2.45 ^g^	42.2 ± 1.53 ^d^	74.57 ± 1.05 ^d^
JP704	46.34 ± 3.05 ^c^	77.17 ± 0.90 ^b^	53.06 ± 1.84 ^c^	64.27 ± 1.04 ^d^	49.62 ± 1.81 ^c^	81.22 ± 0.52 ^b^
JP705	42.40 ± 1.81 ^d^	73.3 ± 0.70 ^c^	41.23 ± 0.85 ^e^	72.42 ± 0.15 ^c^	42.53 ± 1.71 ^d^	74.25 ± 0.42 ^d^
JP710	51.21 ± 2.31 ^b^	67.6 ± 1.16 ^d^	50.53 ± 1.25 ^d^	76.78 ± 0.71 ^b^	51.25 ± 1.36 ^bc^	77.32 ± 0.82 ^c^
JP718	25.72 ± 1.07 ^e^	27.7 ± 2.40 ^f^	23.01 ± 1.60 ^f^	40.14 ± 1.85 ^e^	23.89 ± 1.31 ^e^	44.26 ± 1.61 ^f^
JP1502	62.58 ± 1.40 ^a^	68.2 ± 1.22 ^d^	66.40 ± 1.16 ^a^	80.51 ± 0.48 ^a^	62.03 ± 0.67 ^a^	83.12 ± 0.44 ^a^
JP1510	52.03 ± 1.18 ^b^	77.51 ± 0.52 ^b^	51.99 ± 1.42 ^cd^	62.19 ± 0.92 ^d^	52.81 ± 1.38 ^b^	83.35 ± 0.54 ^a^

Note: Values are the mean ± standard deviation of triplicates. ^a–g^ Mean values in same column with different lowercase letters differ significantly (*p* < 0.05).

**Table 4 foods-11-03476-t004:** Auto-aggregation (%) and Co-aggregation (%) of LAB strains.

Strain	Auto-Aggregation (%)	Co-Aggregation (%)		
		*E. coli*	*S. typhi*	*Staph. aureus*
JP001	8.76 ± 0.60 ^ef^	22.14 ± 1.81 ^c^	21.00 ± 1.43 ^c^	41.41 ± 1.24 ^d^
JP003	13.87 ± 2.21 ^d^	63.94 ± 1.17 ^a^	28.22 ± 0.40 ^b^	77.61 ± 0.61 ^a^
JP701	6.52 ± 0.56 ^f^	7.09 ± 4.25 ^f^	19.24 ± 1.43 ^d^	35.96 ± 4.39 ^e^
JP702	9.88 ± 0.42 ^e^	11.64 ± 2.77 ^e^	15.12 ± 0.99 ^f^	51.21 ± 1.44 ^c^
JP703	7.65 ± 0.13 ^f^	11.29 ± 1.69 ^e^	16.78 ± 0.08 ^e^	14.74 ± 1.61 ^g^
JP704	17.61 ± 1.11 ^b^	13.12 ± 2.03 ^de^	14.87 ± 0.65 ^f^	26.55 ± 1.24 ^f^
JP705	7.57 ± 0.42 ^f^	14.75 ± 0.05 ^de^	13.02 ± 0.40 ^g^	16.12 ± 2.99 ^g^
JP710	2.11 ± 0.55 ^g^	24.95 ± 1.61 ^c^	14.34 ± 0.68 ^fg^	26.13 ± 2.00 ^f^
JP718	10.20 ± 0.18 ^e^	15.37 ± 1.45 ^d^	15.52 ± 0.82 ^ef^	58.19 ± 1.81 ^b^
JP1502	15.92 ± 0.45 ^c^	40.23 ± 1.10 ^b^	38.78 ± 0.61 ^a^	50.22 ± 0.79 ^c^
JP1510	27.77 ± 0.83 ^a^	23.34 ± 1.34 ^c^	12.57 ± 1.30 ^g^	16.09 ± 5.49 ^g^

Note: Values are the mean ± standard deviation of triplicates. ^a–g^ Mean values in same column with different lowercase letters differ significantly (*p* < 0.05).

**Table 5 foods-11-03476-t005:** Hydrophobicity percentage of LAB strains toward different solvents.

Strain	Hydrophobicity (%)		
	Chloroform	Ethyl Acetate	Hexane
JP001	14.43 ± 0.97 ^f^	70.31 ± 0.43 ^a^	52.17 ± 0.91 ^b^
JP003	37.80 ± 1.12 ^c^	45.30 ± 0.57 ^e^	23.75 ± 2.03 ^e^
JP701	16.09 ± 0.55 ^f^	15.71 ± 2.51 ^g^	21.21 ± 1.28 ^e^
JP702	26.90 ± 0.91 ^d^	31.02 ± 2.44 ^f^	19.66 ± 1.24 ^e^
JP703	21.39 ± 1.93 ^e^	58.80 ± 2.26 ^c^	88.44 ± 0.34 ^a^
JP704	38.55 ± 1.49 ^c^	44.49 ± 2.19 ^e^	8.01 ± 4.47 ^f^
JP705	43.07 ± 0.67 ^bc^	58.81 ± 2.55 ^c^	86.98 ± 0.47 ^a^
JP710	30.27 ± 2.91 ^d^	51.55 ± 3.69 ^d^	8.44 ± 4.38 ^f^
JP718	44.60 ± 3.51 ^b^	63.53 ± 2.64 ^b^	6.71 ± 3.42 ^f^
JP1502	74.62 ± 4.38 ^a^	64.04 ± 0.55 ^b^	47.11 ± 1.53 ^c^
JP1510	40.16 ± 1.50 ^c^	19.04 ± 3.28 ^g^	28.10 ± 2.21 ^d^

Note: Values are the mean ± standard deviation of triplicates. ^a–g^ Mean values in the same column with different lowercase letters differ significantly (*p* < 0.05).

**Table 6 foods-11-03476-t006:** Hemolytic activity and antibiotic susceptibility of LAB strains.

Strain	Hemolytic Activity	Antibiotic Susceptibility									
		C	VAN	S	PEN	CC	PIP	OX	CZ	SXT	MT
JP001	−	S	R	S	R	R	R	S	R	S	S
JP003	−	S	R	S	R	S	S	R	R	S	S
JP701	−	S	R	S	S	R	R	R	S	S	S
JP702	−	S	R	S	S	S	S	S	S	R	S
JP703	−	S	R	S	S	S	R	R	S	S	S
JP704	−	S	S	S	S	S	S	S	S	S	S
JP705	−	S	R	S	S	S	R	R	S	S	S
JP710	−	S	R	S	R	S	S	S	S	S	S
JP718	−	S	S	S	S	S	S	S	S	S	S
JP1502	−	S	R	S	S	S	S	S	R	R	S
JP1510	−	S	R	S	S	S	S	S	R	R	S

Note: “−”: No hemolytic activity. R: Resistance, S: Susceptible, C: Chloramphenicol (30 μg), VAN: vancomycin (30 μg), S: streptomycin (10 μg), PEN: penicillin (10 μg), CC: clindamycin (2 μg), PIP: piperacillin (100 μg), OX: oxacillin (1 μg), CZ: cefazolin (30 μg), SXT: trimethoprim-sulfamethoxazole (25 μg), MT: minocycline (30 μg).

**Table 7 foods-11-03476-t007:** Antibacterial activity of LAB strains.

Strain	Antibacterial Activity		
	*E. coli*	*S. typhi*	*Staph. aureus*
JP001	++	++	++
JP003	+	++	+
JP701	+	+++	++
JP702	++	+++	+
JP703	+	+	++
JP704	++	++	++
JP705	++	+	+
JP710	+	+++	+
JP718	+	++	+
JP1502	+	++	++
JP1510	++	+++	++

Note: Interpretation of the diameter of the inhibition zone. “+”: >7.8 mm, <10 mm,”++”: >10 mm, <15 mm, “+++”: >15 mm.

**Table 8 foods-11-03476-t008:** Hypoglycemic ability of LAB strains.

Strain	α-Amylase Inhibition Rate (%)		α-Glucosidase Inhibition Rate (%)	
	CFS	IC	CFS	IC
JP001	44.71 ± 2.97 ^e^	66.83 ± 0.36 ^b^	49.63 ± 0.37 ^e^	43.96 ± 0.43 ^f^
JP003	17.19 ± 2.4 ^g^	72.38 ± 1.07 ^a^	61.74 ± 0.59 ^a^	46.38 ± 0.20 ^e^
JP701	86.15 ± 1.91 ^a^	58.00 ± 0.91 ^c^	51.19 ± 0.45 ^d^	46.54 ± 0.29 ^e^
JP702	84.58 ± 3.76 ^a^	59.86 ± 0.89 ^c^	53.24 ± 0.78 ^c^	47.88 ± 0.22 ^c^
JP703	58.31 ± 3.20 ^d^	45.36 ± 1.20 ^e^	60.86 ± 0.18 ^a^	46.41 ± 0.37 ^e^
JP704	75.94 ± 2.89 ^b^	41.50 ± 1.16 ^f^	49.38 ± 0.98 ^e^	48.53 ± 0.31 ^b^
JP705	26.16 ± 1.88 ^f^	43.15 ± 1.56 ^ef^	59.67 ± 0.40 ^b^	48.56 ± 0.49 ^b^
JP710	47.88 ± 2.71 ^e^	40.43 ± 1.65 ^f^	51.29 ± 0.25 ^d^	45.94 ± 0.17 ^e^
JP718	65.99 ± 2.73 ^c^	37.16 ± 2.02 ^g^	59.45 ± 0.53 ^b^	54.61 ± 0.42 ^a^
JP1502	64.14 ± 3.91 ^c^	29.11 ± 1.52 ^h^	45.31 ± 0.63 ^g^	47.16 ± 0.49 ^d^
JP1510	62.72 ± 2.29 ^cd^	51.26 ± 1.50 ^d^	48.17 ± 0.24 ^f^	44.34 ± 0.35 ^f^
Acarbose	58.02 ± 0.56		74.23 ± 0.09	

Note: Values are the mean ± standard deviation of triplicates. ^a–h^ Mean values in same column with different lowercase letters differ significantly (*p* < 0.05).

**Table 9 foods-11-03476-t009:** Identification of 11 isolates by 16S rRNA gene sequencing and their accession numbers from GenBank.

Strain	Species	NCBI Accession No.
JP001	*Weissella cibaria*	OP288150
JP003	*Pediococcus acidilactici*	OP288151
JP701	*Lactiplantibacillus plantarum* subsp. *plantarum*	OP288152
JP702	*Lentilactobacillus parabuchneri*	OP288153
JP703	*Pediococcus pentosaceus*	OP288154
JP704	*Levilactobacillus brevis*	OP288155
JP705	*Pediococcus pentosaceus*	OP288156
JP710	*Leuconostoc mesenteroides*	OP288157
JP718	*Levilactobacillus brevis*	OP288158
JP1502	*Lentilactobacillus parabuchneri*	OP288159
JP1510	*Lentilactobacillus parabuchneri*	OP288160

## Data Availability

The data presented in this study are available within the article.

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
