# Peer review of "Probiotic Properties Including the Antioxidant and Hypoglycemic Ability of Lactic Acid Bacteria from Fermented Grains of Chinese Baijiu"

_foods, 2022, doi:10.3390/foods11213476_

Round 1

Reviewer 1 Report

Dear Authors,

I write you in regard to the manuscript entitled Probiotic properties including the antioxidant and hypoglycemic ability of Lactic acid bacteria form fermented grains of Chinese Baijiu.

- please, if possible, give some example in lines 33/34.

- in line, 53, please, explain what  would be "excellent" probiotics. Or suppress the term.

- please, add some discussion in 3.1.

- please, consider revising the text in lines 209-218.

- please, give some example in line 284.

- please, add more photographs in 3.6.

- please, eliminate all speculation in Conclusions.

Overall, it is suggested to discuss in a deeper way the results. 

Reviewer 2 Report

English language and style must be improved. The article is not prepared in accordance with the journal's editorial requirements. There are some comments:

·       All abbreviations should be defined when used for the first time, also in the Abstract.

·       Line 7: correct the following: “…the probiotic properties of probiotics…”

·       Line 16: what does it mean “the best”?

·       The purpose of the work is chaotic and incomprehensible.

·       Lines 27-28: probiotics do not created. Correct the sentence.

·       There are so many typos in the manuscript… it all must be corrected.

·       Section 2.2. What was the aim to prepare CFSs?

·       Line 72: you should give authors surname et al. additionally to the citation number.

·       What is the novelty of the study? Describe.

·       Line 76: I do not understand “The strains generated by thallus precipitation were selected”.

·       Latin names of microorganisms must be written in italics.

·       Line 123: what strain of S. aureus?

·       What is Salmonella typhi? Give correct name.

·       Lines 148-149: what groups?

·       Table 1 caption says nothing. In the Table what does it mean plus and what minus??

·       The headings in the tables are different, different fonts are used, bold or not, lowercase or uppercase. This needs to be standardized.

·       Figure 1 caption says nothing. What does it mean partial. Explain what is in each photo.

·       Line 350: 16S RNA?

·       Authors should apply new nomenclature for Lactobacillus species, as it was divided into 25 genera. See: LACTOTAX webpage http://lactotax.embl.de/wuyts/lactotax/ and

Zheng J, Wittouck S, Salvetti E, et al. A taxonomic note on the genus Lactobacillus: Description of 23 novel genera, emended description of the genus Lactobacillus Beijerinck 1901, and union of Lactobacillaceae and Leuconostocaceae. Int J Syst Evol Microbiol. 2020. doi: 10.1099/ijsem.0.004107.

·       Captions to tables and figures say nothing.

·       There is poor discussion.

·       Probiotics can be called a strain when it has undergone in vivo testing and shown some properties. In vitro tests are not probiotics. So isolated strains are not probiotics. Please take this into account when improving the manuscript.

Round 2

Reviewer 1 Report

Dear Authors,

The new version of your manuscript was clearly improved. 

Reviewer 2 Report

The article was modified.